# Valorization of Carob Fruit Residues for the Preparation of Novel Bi-Functional Polyphenolic Coating for Food Packaging Applications

**DOI:** 10.3390/molecules24173162

**Published:** 2019-08-30

**Authors:** Vlasios Goulas, Loukas Hadjivasileiou, Alexandra Primikyri, Christodoulos Michael, George Botsaris, Andreas G. Tzakos, Ioannis P. Gerothanassis

**Affiliations:** 1Department of Agricultural Sciences, Biotechnology and Food Science, Cyprus University of Technology, Limassol 3603, Cyprus; 2Department of Chemistry, University of Ioannina, Ioannina GR-45110, Greece

**Keywords:** carob fruit, antioxidant activity, antimicrobial activity, salmon, coating, polymerization, polyphenols

## Abstract

The food industry has become interested in the development of innovative biomaterials with antioxidant and antimicrobial properties. Although several biopolymers have been evaluated for food packaging, the use of polyphenolic coatings has been unexplored. The purpose of this work was to develop an antioxidant and antimicrobial coating for food packaging through the polymerization of carob phenolics. At first, the polyphenolic coatings were deposited in glass surfaces polymerizing different concentrations of carob extracts (2 and 4 mg mL^−1^) at three pH values (7, 8 and 9). Results demonstrated that the coating produced at pH 8 and at a concentration of 4 mg mL^−1^ had the most potent antioxidant and antimicrobial potential. Then, the coating was applied directly on the salmon fillet (coating) and on the plastic container (active packaging). Peroxide and thiobarbituric acid-reactive substances (TBARS) methods were used to measure the potency to inhibit lipid oxidation in salmon fillets. Furthermore, the anti-Listeria activity of coatings was also assessed. Results showed a significant decrease of lipid oxidation during cold storage of salmon fillets for both treatments; the superiority of applied coating directly on the salmon fillets was also highlighted. Regarding the antimicrobial potency, the polyphenolic coating depleted the growth of *Listeria monocytogenes* after 10 days storage; while the active packaging had no effect on *Listeria monocytogenes*. Overall, we describe the use of low-cost carob polyphenols as precursors for the formation of bifunctional coatings with promising applications in food packaging.

## 1. Introduction

Food packaging stands as a challenge for food manufactures since it has to fulfil the need to protect the quality of food, promote usability, provide information and communicate with consumers. It is common sense that food packaging is one of the most important factors for consumer acceptance [1,2]. At the same time, the development of new packaging materials requires a consideration from social, environmental and economic points of view as the production and use of these materials generates bulky volumes of waste and consumes raw materials, water and energy [3]. Taking into consideration the above, researchers and food industries pay attention to develop alternative food packaging materials. In this attempt, the design and production of packaging materials from food grade or underutilized food ingredients has been adopted. Furthermore, the utilization of food industry wastes has also been espoused for the preparation food packaging materials [4]. More specific, biopolymers such as polysaccharides, proteins, lipids and gums are used to produce “green” and “sustainable” packaging materials utilizing raw ingredients from marine and agricultural sources [5]. Only in the last decade, numerus studies and patents have been published related to edible coatings. However, there has been no attempt to utilize natural phenols to produce coatings for food packaging; they are only exploited as active ingredients in biopolymer coatings [6].

Recently, Sileika and co-workers described the preparation of multifunctional coatings through the polymerization of plant polyphenols at alkaline pH [7]. They reported the potency of dihydroxyphenol and trihydroxyphenol compounds to formulate coatings inspired by the polymerization of catecholamines in mussels for the production of polydopamine. Subsequently, many studies utilized pure phenolic compounds and plant extracts to produce polyphenolic coatings that are mainly exploited in the fabrication of functionalized material with antimicrobial, adhesive, anti-corrosion and anti-fouling properties [8,9,10] or to act as photoinitiating, transparent thermoresponsive superhydrophobic surfaces [11,12,13]. 

The carob fruit is used by the food industry for the production of locust bean gum, which is well-known as food additive E410. The carob pulp is also utilized for the production of carob syrup and molasses through the recovery of sugars. Although the residual parts of carob fruits are rich in natural phenols, they are discarded or used as animal feed [14]. Taking into consideration the abundance of phenolic compounds in carob fruits, the objective of the present work was to valorize the carob phenolic fraction to develop an antioxidant and antimicrobial coating for food packaging. 

## 2. Results

### 2.1. Preparation of Coatings

Based on previous studies, the formation of polyphenolic coatings was performed via polymerization of carob phenolics at pH values 7, 8 and 9 under mild agitation for 24 h. Two concentrations of carob extract were also tested. Carob extract can be considered as an ideal source of di- and tri-hydroxyphenols; gallic acid, catechins and gallotannins are the main constituents of the phenolic fraction in carob fruits [14]. These phenolics are molecular precursors for the formation of coatings as concluded by Barrett et al. who compared a library of 20 natural and synthetic polyphenols [8]. The polymerization of carob extract phenolics in alkaline saline was directly observed as the color of the reaction solution changed from slight yellow to brown. The deposition of the polymerized phenolic coatings onto the surface of slides was also confirmed by the formation of a brown coating after 24 h. The generation of polyphenolic coatings was further identified with the employment of NMR spectroscopy. Figure 1A illustrates a selected region of the ^1^H-NMR spectrum of the carob extract which shows numerous resonances in the aromatic region due to the presence of phenolic compounds. Of particular interest is the presence of a singlet at 7.00 ppm which should be attributed to gallic acid since this resonance increases upon spiking with gallic acid (Figure 1B). These resonances disappeared after polymerization under alkaline saline conditions (Figure 1C). The latter occurs as the carob phenolics contain least one aromatic vicinal diol ring, which is required to produce phenol bioinspired coatings. 

### 2.2. In Vitro Assessment of Polyphenolic Coatings

At first, the phenolic content and antioxidant effect of coated slides were evaluated. Results showed a significant impact of pH value on phenolic contents of coatings (Figure 2A). On the contrary, the concentration of carob extract had no effect on their phenolic contents. More specific, the total phenolic groups of coated slides ranged from 102.3 mg gallic acid equivalents (GAE) cm^−2^ to 17,953.2 mg GAE cm^−2^. Similarly, the antioxidant activity of coated slides was mainly affected by alkaline conditions of polymerization (Figure 2B). The deposition of coating at pH 8 resulted in the most active coated slides. It is noteworthy that the differences in antioxidant activity are lower than the differences in phenolic group contents; the DPPH values fluctuated between 9.3 nmol 6-hydroxy-2,5,7,8-tetramethylchroman-2-carboxylic acid (Trolox) equivalents cm^−2^ and 30.6 nmol Trolox equivalents cm^−2^. The ability of coated slides to act as radical scavengers is mainly ascribed to the catechol and gallol groups of coatings. These groups can implement as hydrogen-atom donors due to the extra stability of phenoxy radicals that stems from their hydrogen-bonding interactions with adjacent hydroxy groups. Furthermore, the gallol groups undergo partial electron transfer processes while reducing DPPH radicals [15]. 

Afterwards, the antimicrobial potency of coated slides against *Listeria monocytogenes* ATCC23074 was assessed. Results showed a similar trend with total phenolic content and antioxidant activity. The pH value is a critical parameter for anti-Listeria potency of coated slides. The most promising coatings were prepared at pH 8, followed by the formation of coatings at pH 9 and pH 7 (Table 1). Previous studies also stated that the polyphenolic coatings exert significant growth inhibition against Gram-positive bacteria such as *Staphylococcus aureus* [7,9]; no data are available for the potency of polyphenolic coatings to inhibit the growth of Listeria bacteria. The antimicrobial activity of monomeric phenolics is well-established. Its activity is linked with a variety of mechanisms such as membrane interaction, DNA gyrase inhibition, metal sequestering, enzyme inhibition and reactive oxygen generation [16]. The mechanism of antimicrobial action of natural phenolic polymers has been less studied. However, these phenolic polymers can reduce iron intake, due to its chelating properties, leading to a modulation of cellular oxidative stress. Bacterial swarming motility and biofilm formation can also be affected by their presence. This could possibly be due to iron ions depletion or to direct binding of polymers to cell surface structures including lipo-polysaccharides and flagellin [17].

Overall, the preparation of bi-functional polyphenolic coating for food packaging applications at a pH value of 8 was preferred due to its antioxidant and antimicrobial properties. Although the concentration of carob extract had no significant effect on the antioxidant and antimicrobial properties of coatings, the high concentration (4 mg mL^−1^) was preferred for the preparation of coatings for salmon fillets.

### 2.3. In Vivo Assessment of Polyphenolic Coatings

The potential of polyphenolic coatings to inhibit the lipid oxidation in salmon was studied with the employment of two assays, namely peroxide and thiobarbituric acid-reactive substances (TBARS) values. The peroxide value is an essential index for lipid quality. It provides an estimation of the overall oxidation status for lipid-containing foods in the primary phase of oxidation, generally known as the induction period. Figure 3A demonstrates a significant inhibition of polyphenolic coating on the peroxide value of packed salmon fillets for both treatments. More specifically, a reduction of about 20%–25% in lipid oxidation was found after 5 and 10 days storage at 6 °C. The decline of the peroxide value with the application of polyphenolic coating in salmon is comparable with the use of gelatin coating, a well-known biopolymer [18]. Subsequently, the TBARS of packed salmon during cold storage were determined since this assay quantifies the secondary products of lipid oxidation that are responsible for off-flavors/odors [19]. Results showed a gradual increase of TBARS values for all treatments; whereas the active packaging and coating retarded the rate of oxidation. TBARS values of uncoated salmon fillets came up to the critical limit of 2 mg kg^−1^ that is correlated with the negative impact of stored salmon [20]. On the contrary, the polyphenolic coating kept the TBARS values lower than 1.4 mg kg^−1^ avoiding the presence of undesirable off-flavors. Figure 3B clearly states that the utilization of polymerized polyphenols as coating is superior to its use as a component of active packaging. The coating decreased the formation of TBARS approximately 40% after 10 days of cold storage, while active packaging achieved a reduction of 28%. Overall, the coating is more appropriate than active packaging due to its more uniform contact to the surface of salmon filltes. Similar findings were also reported for the application of xanthan film and coating to inhibit lipid oxidation in smoked salmon [21].

The potency of polyphenolic coating to prevent the growth of *Listeria monocytogenes* ATCC23074 on salmon fillets was also investigated. This facultative anaerobic bacterium was spiked onto the surface of salmon fillets since *Listeria monocytogenes* contamination is one of the significant concerns of raw salmon [22]. Our results highlighted the potency of polyphenolic coating to act as an anti-Listeria agent (Figure 4). The use of polymerized solution to form a coating for salmon fillet significantly decreased the growth of *Listeria monocytogenes*, although an unrealistic high contamination was applied on the surface of salmon. Surprising, a reduction of bacterial growth was observed for all treatments, even the control samples after six days of cold storage. The latter is explained by the fact that it will take hours for the Listeria cells to divide and not all of the cells will make it through the process at low temperature. In particular, one log reduction of *Listeria monocytogenes* was found after 10 days storage. On the other hand, the use of polymerized carob phenols as active packaging had no significant antimicrobial activity. As mentioned above, the rough surface of salmon is overlaid better using coating than active packaging.

## 3. Materials and Methods 

### 3.1. Chemicals and Carob Products

Carob fruits (*Ceratonia siliqua* L., cvs ‘Tilliria’) were harvested at maturity stage from an experimental orchard (Avdimou, Limassol district, Cyprus), based on size uniformity and external color. At first, the carob fruits were grinded using an electric stainless-steel coffee grinder (Bestron AKM1405 150W, Bestron Nederland BV, s-Hertogenbosch, the Netherlands). Then, approximately 1 kg of ground carob fruit was mixed with 3 L of water for 24 h in order to recover sugars as it is applied for the preparation of carob syrup. The filtrate (sugar fraction) was discarded and the residue was dried at 50 °C until constant weight. The dried residue (6 g 100 mL^−1^) was extracted with methanol to obtain phenols for 30 min at 60 °C in an ultrasonic bath (UCI-50, 35 kHz, Raypa-R. Espinar, S.L., Terrassa, Barcelona, Spain). After extraction, the mixture was filtrated, and the methanol was removed with the employment of a rotary evaporator in order to obtain carob extract. This extract was used to prepare bi-functional polyphenolic coatings. The preparation of carob extract is illustrated in Figure 5.

### 3.2. Preparation Coatings at Different pH Values and at Different Concentrations of Extract

The deposition of polyphenolic coatings on the substrates was performed as previous studies with slight modifications [7,8]. In particular, the microscope slides (12.5 × 38 mm) were immersed into 2 mg mL^−1^ and 4 mg mL^−1^ solutions of carob extracts at pH values 7, 8 and 9 for 24 h at 25 °C with mild agitation on an incubator shaker. All buffers were prepared at 100 mM buffer and 600 mM NaCl. The buffers used were as follows: bis-Tris (pH 7) and bicine (pH 8 and 9). Then, the slides were removed from solution, rinsed with deionized water and dried.

### 3.3. Screening of Carob Phenolic Polymerization by Nuclear Magnetic Resonance (NMR) Spectroscopy

NMR experiments were performed on a Bruker AV500 spectrometer equipped with a BBI probe (Bruker Biospin, Rheinstetten, Germany). Samples were dissolved in 600 μL of deuterium oxide and transferred to 5 mm NMR tubes. The NMR spectrometer was controlled by the software TopSpin 2.1. One-dimensional (1D) ^1^H-NMR spectra were acquired with an acquisition time of 2.72 s, relaxation delay of 2 s, spectral width of 12 ppm and 90° pulse length. Spiking of gallic acid was performed in a solution of the extract in deuterium oxide by adding a solution of 0.6 mM gallic acid in chloroform-*d*.

### 3.4. Determination of Total Phenol Content and Antioxidant Potency of Coatings

The phenol content of nine coated slides for each treatment was quantified using a Folin−Ciocalteu assay adapted for active packaging film [23]. More specifically, each slide was submerged in 1 mL of 0.2 N Folin−Ciocalteu reagent and 0.2 mL of 0.05 N HCl and incubated with shaking for 5 min. Afterwards, 0.8 mL of 7.5% sodium carbonate was added to the reaction mixture and incubated for 2 h with shaking in the dark. The absorbance of the reaction solution was measured at 760 nm. A standard curve of gallic acid in 0.05 N HCl was prepared and results are expressed as mg GAE cm^−2^.

The antioxidant activity of nine coated slides for each treatment was determined by DPPH (1, 1-diphenyl-2-picrylhydrazyl) colorimetric assay as adapted for coatings [7]. Each slide was immersed in 2.5 mL DPPH 30 μΜ and 2.5 mL methanol. After an incubation for 30 min, the absorbance of the mixture was measured at 517 nm. A standard curve of Trolox was prepared and results are expressed as nmol Trolox equivalents cm^−2^.

### 3.5. Determination of Anti-Listeria Activity of Coatings

The anti-Listeria activity of the coating’s surface was determined according to Nithya and co-workers [24]. Briefly, nine coated slides (12.5 × 38 mm) for each treatment were placed onto the surface of BHI soft grown indicator organism. The treated face of the coating was agar plates seeded with 100 μL *Listeria monocytogenes* ATCC23074 10^6^ cfu mL^−1^ in contact with the agar. The untreated films were also assayed and served as negative controls. After incubating at 37 °C for 5 days, the efficiency of coatings was evaluated. The antimicrobial activity of the coated surfaces was assigned as slight antimicrobial activity (+), moderate antimicrobial activity (++) and high antimicrobial activity (+++) based on the density of sessile growth.

### 3.6. Application of Carob Polyphenolic Coating on the Salmon

Fresh salmon (*Salmo salar*) was purchased from a local supermarket. The crude chemical analysis of salmon was 25.5 g fat 100 g^−1^, 17.2 g protein 100 g^−1^ and 0.2 g carbohydrates 100 g^−1^. The fresh salmon was parceled out in slices of 25 g for each measurement (peroxide value, TBARS assay, Listeria growth). Then, the surface of each slice was treated under UV light for 10 min and then, 10 μL *L. monocytogenes* 10^6^ cfu mL^−1^ were inoculated on the surface of salmon samples and air-dried for 30 min. Subsequently, the salmon was used to apply carob polyphenolic coating. At first, the salmon slices were dipped into polymerized solution for 5 min, drained, air-dried at room temperature, vacuum-packed and subsequently cold-stored at 6 °C for 10 days (coating). At the second treatment, the polyphenolic coating was applied on the interior of oriented polyamide/polyethylene (OPA/PE) bags. More specifically, 1 mL of polymeric solution was spread onto a plastic bag and allowed to air-dry for treatments; the control samples were vacuum-packed into OPA/PE bags without polyphenolic coating. Then, the inoculated salmon slices were vacuum-packed and stored at 6 °C for 10 days (active packaging). Similarly, the coatings were applied on salmon to study their potency to prevent lipid oxidation.

### 3.7. Measurement the Inhibition of Lipid Oxidation by TBARS and Peroxide Value Assays

Peroxide value was determined according to our previous study [19]. A mass of 5 g of salmon was weighed into a 250-mL Erlenmeyer flask, followed by the addition of 30 mL of the mixture of acetic acid and chloroform (3:2, *v/v*) and 0.5 mL of saturated potassium iodate solution. The mixture was allowed to react for 1 min, then 30 mL of deionized water and 1 mL of starch solution (10 g L^−1^) were added. The mixture was titrated with 0.01 M Na_2_S_2_O_3_ until the blue color disappeared. All the samples were repeated for six replicates. Results were expressed in meq kg^−1^ fish.

The lipid oxidation was also determined by quantification of secondary lipid oxidation products using the thiobarbituric acid-reactive substances (TBARS) assay. A total of 5 g of salmon was extracted with 30 mL of trichloroacetic acid 75 g L^−1^. The solution was filtered and aliquots of 5 mL from the filtrate were reacted with 5 mL of 2-thiobarbituric acid 0.02 M at 95 °C for 40 min. Then, the mixture was cooled and measured at 538 nm. A standard curve was prepared using 1,1,3,3-tetramethoxypropane as standard and results were expressed as mg malondialdehyde (MDA) kg^−1^ fish [19]. All the samples were repeated for six replicates.

### 3.8. Measurement the Inhibition of Listeria Growth

Plating and enumeration of the *Listeria monocytogenes* ATCC23074 was performed using a modification of the method described by Botsaris and Taki [25]. Briefly, from each treatment, 25 g were aseptically weighed and transferred into sterile stomacher bags, where they were mixed with 225 mL of sterile maximum recovery diluent (MRD; Thermo Fischer Scientific Oxoid, Basingstoke, UK) and homogenized in a stomacher circulator for 30 s. Serial dilutions (1:10) were prepared using the MRD down to 10–12. A volume of 0.1 mL of each dilution was inoculated and dispersed onto AL agar (Agar Listeria according to Ottaviani and Agosti, BioRad) in duplicate and incubated at 37 °C for 3 days. Results were expressed as cfu g^−1^.

### 3.9. Statistical Analysis

Statistical analysis was carried out using the software package SPSS v22.0 (SPSS Inc., Chicago, IL, USA) and the comparison of averages of each treatment was based on the analysis of variance (one-way ANOVA) according to Duncan’s multiple range test at 5% significance level. 

## 4. Conclusions

The formation of polyphenolic coatings via in situ oxidative polymerization at alkaline pH can be considered as an alternative strategy to produce sustainable packaging materials for foods. The present study describes the valorization of carob fruit wastes to prepare a bifunctional coating to retain the quality of salmon. Our findings showed that the polyphenols can produce antioxidant and antimicrobial materials for food packaging. Optimization of the preparation and application of polyphenolic coatings will further improve its antioxidant and antimicrobial potency as it is the first attempt to exploit for food packaging. In conclusion, the present study opens new horizons for the use of green polyphenolic coatings in food packaging as these coatings have been only utilized for the production materials of medical and mechanical interest.

## Figures and Tables

**Figure 1 molecules-24-03162-f001:**
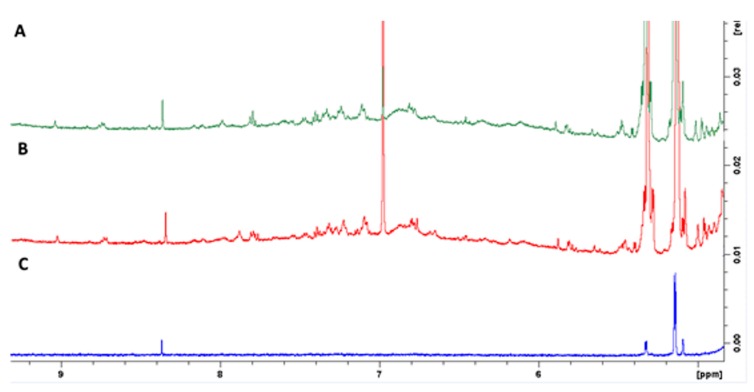
Selected region of the ^1^H-NMR spectra (500 MHz) of: (**A**) carob extract, (**B**) the same extract as in (**A**) after spiking with gallic acid, and (**C**) coating.

**Figure 2 molecules-24-03162-f002:**
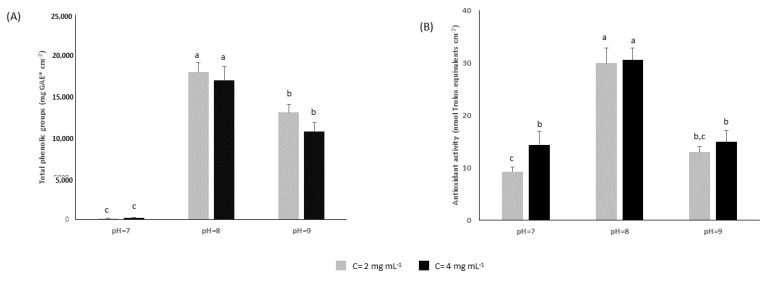
Assessment of (**A**) total phenolic groups and (**B**) antioxidant activity of polyphenolic coated slides. Values followed by the same letter do not differ significantly according to the least significant difference test (LSD, *p* = 0.05).

**Figure 3 molecules-24-03162-f003:**
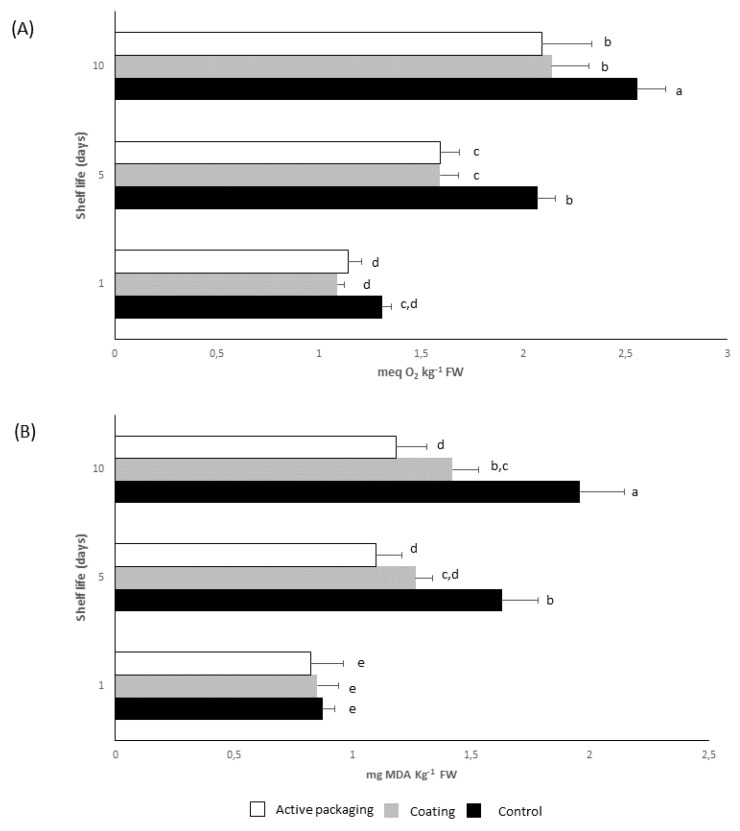
Antioxidant effect of polyphenolic coating on salmon fillets as measured by: (**A**) peroxide value, and (**B**) thiobarbituric acid-reactive substances (TBARS) value stored at 6 °C for 10 days. Values followed by the same letter do not differ significantly according to the least significant difference test (LSD, *p* = 0.05).

**Figure 4 molecules-24-03162-f004:**
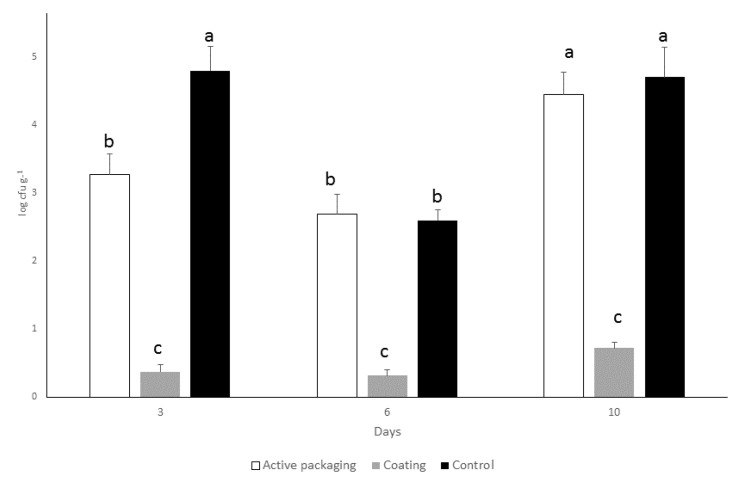
Anti-Listeria effect of polyphenolic coating on salmon fillets stored at 6 °C for 10 days. Values followed by the same letter do not differ significantly according to the least significant difference test (LSD, *p* = 0.05).

**Figure 5 molecules-24-03162-f005:**
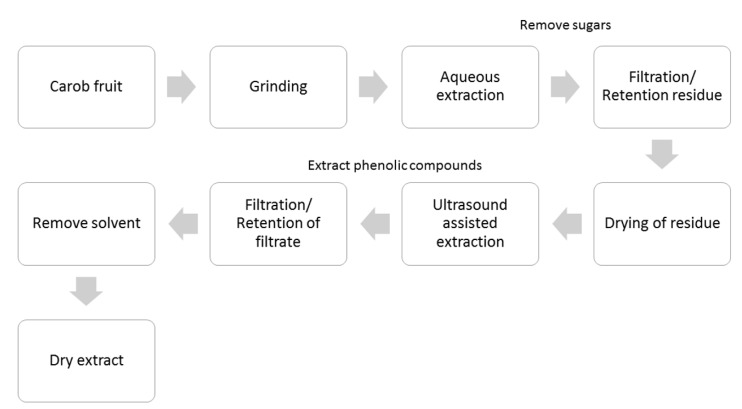
Schematic representation of preparation of the phenolic fraction from carob fruit.

**Table 1 molecules-24-03162-t001:** Anti-Listeria activity of coated slides after incubation for 10 days at 6 °C. The activity is denoted as weak (+), moderate (++) and strong (+++).

Coating	Anti-Listeria Activity	Coated Slides
Control	−	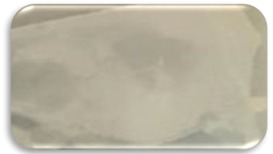
pH 7 (2 mg mL^−1^)	+	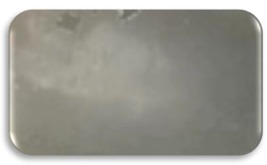
pH 7 (4 mg mL^−1^)	+	
pH 8 (2 mg mL^−1^)	+++	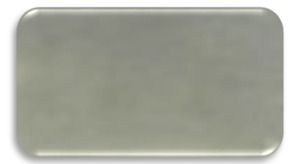
pH 8 (4 mg mL^−1^)	+++	
pH 9 (2 mg mL^−1^)	++	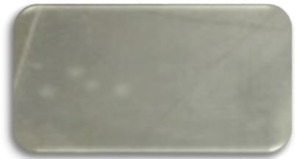
pH 9 (4 mg mL^−1^)	++	

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
