# Peer review of "Valorization of Carob Fruit Residues for the Preparation of Novel Bi-Functional Polyphenolic Coating for Food Packaging Applications"

_molecules, 2019, doi:10.3390/molecules24173162_

Round 1
Reviewer 1 Report
The manuscript deals with the valorization of carob fruit residues for the preparation of novel bi-functional polyphenolic coating and their potential applications in food packaging (coating). Paper is well written and in scope of the journal. Standard methodologies were used in the paper and interesting results are presented.
My major critics is related to Materials and methods and some Result (discussion), it is not well written, and discussion should be improved in some parts.
Line 64-240: Did you prepare only one extract of carob fruits and how many replicated did you do in all assays (phenols, antimicrobial tests TBARS . . .)
Line 2013-2014: How did you decided the antimicrobial activity of coated surfaces was slight (+), moderate (++) or high antimicrobial activity (+ + +)? Did you measure diameter around coated slides where was no growth of bacteria or . . . ? Please add some sentence about these.
Line 237: I suppose that was 95 °C not 950 °C
For determination of anti-Listeria activity of coatings incubation was for 5 days and for measurement the inhibition of listeria growth was 3 days- is there special reason for
Figure 4. You did not mention in the discussion the reason for the lower bacterial growth after 5 days of storage compared to the first day, even in the control sample. Is there any explanation for this?
Reviewer 2 Report
The manuscript deals with the valorization of carob fruits residues for the preparation of novel bi-functional polyphenolic coating for food packaging applications.
Please separate values from units, e.g. “37 °C” not “37°C”.
Abstract
Line 14- “The food industry has turn the interest on the development of biomaterials due the increasing awareness of environmental protection associated to plastic wastes. Although, several biopolymers have been evaluated for the food packaging, the use of polyphenolic coatings has been unexplored. The purpose of this work was to develop an antioxidant and antimicrobial coating for food packaging through the polymerization of carob phenolics.”???but plastic was used in this study. No alternative to plastic materials was presented. The abstract must be revised.
Introduction
Line 41- “Taking into consideration the above, researchers and food industries pay attention to develop alternative food packaging in order to minimize the environment impact.”?? This section must be revised. The authors must discuss better the reduction of food wastes (carob residues and others) rather than the reduction of plastics, since in the present study, plastic packaging was used in all tested salmon samples.
Results
Line 95- “9.3 nmol Trolox cm -2 and 30.6 nmol Trolox cm -2”?? equivalents of Trolox?
Line 154- “Our results highlighted the potency of polyphenolic coating to act as anti-Listeria agent for salmon packaging (Figure 4).”??Please add average and standard deviation. Moreover, please add different superscript letters for significant differences.
Materials and methods
Pictures of each fish sample?fish colour?fish pH??wettablility??colour of coating solution??
Line 169- “1 Kg”??Please replace by “1 kg”.
Line 173- “KHz”??Please replace by “kHz”.
Line 217- “The fresh salmon was parcelled out in slices of 25 g.”?? enough for all analyses??5 g for TBARS, 5 g for peroxide value and 25g for Listeria growth??
Line 219- “10 μL L. monocytogenes 10 6 cfu mL -1”???Please present the correct value used.
Line 222- “vacuum-packaged”?? packaging material??
Line 223- “coating was applied on the interior of plastic bag”??different packaging material from control??
Conclusion
Line 260- “In conclusion, the development of “green” polyphenolic coatings for food packaging is a promising research area as these coatings have been used successfully for the production materials of medical and mechanical interest.”?? Please rephrase.
References
Please format the title of each article according to the guide for authors.
Round 2
Reviewer 2 Report
Conclusion
Line 292- “The optimization of preparation and application of coatings may will further improve its performance.”???Please rephrase.
Lone 293- “In conclusion, the present study opens neew horizons for the uses of “green” polyphenolic coatings in the food packaging as these coatings have been only utilized for the production materials of medical and mechanical interest.”???There are several studies in the literature reporting the use of coatings from food wastes. Please rephrase. Also correct typos.
